# Urinary interleukins (IL)-6 and IL-10 in schoolchildren from an area with low prevalence of *Schistosoma haematobium* infections in coastal Kenya

**Kariuki H. Njaanake**[1,2]*, **Job Omondi**[1], **Irene Mwangi**[2], **Walter G. Jaoko**[1,2], **Omu Anzala**[1,2]

1 Department of Medical Microbiology, College of Health Sciences, University of Nairobi, Nairobi, Kenya,
2 KAVI- Institute of Clinical Research, University of Nairobi, Nairobi, Kenya

* kn@uonbi.ac.ke

## Abstract

Urinary cytokines are gaining traction as tools for assessing morbidity in infectious and non-infectious inflammatory diseases of the urogenital tract. However, little is known about the potential of these cytokines in assessing morbidity due to *S. haematobium* infections. Factors that may influence the urinary cytokine levels as morbidity markers also remain unknown. Therefore the objective of the present study was to assess how urinary interleukins (IL-) 6 and 10 are associated with gender, age, *S. haematobium* infections, haematuria and urinary tract pathology and; 2) to assess the effects of urine storage temperatures on the cytokines. This was a cross-sectional study in 2018 involving 245 children aged 5–12 years from a *S. haematobium* endemic area of coastal Kenya. The children were examined for *S. haematobium* infections, urinary tract morbidity, haematuria and urinary cytokines (IL-6 and IL-10). Urine specimens were also stored at –20˚C, 4˚C or 25˚C for 14 days before being assayed for IL6 and IL-10 using ELISA. The overall prevalence of *S. haematobium* infections, urinary tract pathology, haematuria, urinary IL-6 and urinary IL-10 were 36.3%, 35.8%, 14.8%, 59.4% and 80.5%, respectively. There were significant associations between prevalence of urinary IL-6, but not IL-10, and age, *S. haematobium* infection and haematuria ($p$ = 0.045, 0.011 and 0.005, respectively) but not sex or ultrasound-detectable pathology. There were significant differences in IL-6 and IL-10 levels between urine specimens stored at –20˚C and those stored at 4˚C ($p$<0.001) and, between those stored at 4˚C and those stored at 25˚C ($p$<0.001). Urinary IL-6, but not IL-10, was associated with children's age, *S. haematobium* infections and haematuria. However, both urinary IL-6 and IL-10 were not associated with urinary tract morbidity. Both IL-6 and IL-10 were sensitive to urine storage temperatures.

**Data Availability Statement:** The data set analyzed for this manuscript has been provided as supporting information.

**Funding:** This work was solely funded by European and Developing Countries Clinical Trials Partnership (EDCTP) grant number 100440 UCE TMA 2015 CDF - 995 (http://www.edctp.org/projects-2/edctp2-projects/career-development-fellowships/) as post-doctoral support for KHN. No other author received funding from any award, commercial salary, or other funding source. The funder played no role in Study design, Data collection and analysis, Decision to publish, Preparation of the manuscript.

**Competing interests:** The authors have declared that no competing interests exist.

## Introduction

*Schistosoma* spp., the causative agents of human schistosomiasis, infect over 250 million individuals worldwide [1, 2]. Currently, morbidity control through mass praziquantel administration is the main strategy for schistosomiasis control which has been adopted by governments in endemic countries with support from WHO and other stakeholders [3]. This is a long-term resource-intensive strategy and monitoring its performance is imperative. However, most of the currently available tools to monitor schistosomiasis morbidity after treatment with praziquantel have inherent weaknesses including low sensitivity, requirement of high performance freezers and electricity in the field which is a major problem in resource poor endemic areas [4, 5]. Despite this, progress in development of more tools that can circumvent these drawbacks has been slow.

Cytokines in urine are increasingly being appreciated as potential biomarkers of urogenital tract morbidity in other infectious and non-infectious inflammatory conditions [6–11]. Urogenital schistosomiasis is associated with considerable inflammation in the urogenital tract [12]. A study in highly endemic area of coastal Kenya assessed different cytokines and suggested that interleukin (IL)-6 and -10 in urine of children infected with *S. haematobium* are correlated with morbidityand infection intensities [13]. This implied that these cytokines have the potential of being additional tools to monitor *S. haematobium*-related urogenital morbidity. It is known that control activities, including mass praziquantel administration, tend to reduce infection prevalence and intensities in endemic areas which in turn may reduce the performance of morbidity monitoring tools [4]. It is therefore imperative to investigate the potential of urinary IL-6 and IL-10 in morbidity monitoring under low infection endemicity setting. In addition, it is important to assess how various factors may influence these cytokines. For example, age and sex have been shown to be important factors that influence urinary cytokines in other inflammatory conditions [14]. Such information is also important with regard to urinary schistosomiasis but is currently not available.

Schistosomiasis is endemic in low resource settings and monitoring the performance of morbidity control involves collection of specimens in areas with no electricity or freezers for storage. This means that specimens such as urine may require storage for some time before being assayed. A good tool therefore ought to circumvent the requirement of electricity and high performance freezers for specimen storage. In the present study, we therefore also sought to assess how urinary IL-6 and IL-10 are associated with age and sex in children infected with *S. haematobium* in coastal Kenya. In addition, we also assessed the effects of storage of urine specimens at –20˚C, 4˚C and 25˚C for two weeks on urinary IL-6 and IL-10.

## Methods

### Study area and study design

The present cross-sectional study was carried out in the year 2018 in children attending Duncan Ndegwa Primary School. The school is located in Kwale County of coastal Kenya. The County is known to have a low endemicity endemic for *S. haematobium* compared to Tana Delta of Kenya [13, 15]. The study involved 245 randomly selected children whose parents had given informed consent. Only children, boys and girls, aged between 5 and 15 years who were permanent residents of the area were included in the study. These were children who had had no treatment with praziquantel in the past three months and had no observable clinical illness. Demographic information, including age and sex was collected from the children during recruitment. A 50 ml urine sample was collected from each child on each of three consecutive days [16]. A part (10 ml) of day 1 urine sample from each child was examined for *S.*

*haematobium* eggs. Another part of the day 1 urine sample from each child was tested for hae-maturia. A third part was divided into three aliquots which were stored at –20˚C, 4˚C and 25˚C, respectively, for 14 days before being assayed for IL-6 (n = 165) and IL-10 (n = 190). On each of days II and III, a 10 ml urine sample from each child was examined for *S. haematobium* eggs. The urinary tracts of 165 children were also examined for morbidity. children, boys and girls, aged between 5 and 15 years who were permanent residents of the area. Only children who had had no treatment with praziquantel in the past three months and had no observable clinical illness were included in the study.

## Sample size calculation

The minimum sample size for the study was estimated using statistical software (Stata Version 12) as an estimation for a two-sample means comparison using t-test assuming equal standard deviations. The intention was to detect an absolute difference in mean cytokine levels (IL-6 or IL-10) of 0.5 pg/ ml between children with urinary tract morbidity and those without where *p*-value = 0.05; power = 80% and; standard deviation = 1. This gave a minimum sample size of 168 children. However, since absenteeism in the schools was deemed to be high it was assumed that about 45% of the children may not provide all the specimens required. The minimum sample size was therefore increased by 45% to yield 243.6 children. This was rounded off to 245 children.

## Test for haematuria

Immediately after collection, day-1 urine samples were examined visually for macrohaema-turia and for microhaematuria using URiSCAN dipsticks (YD Diagnostics, Korea) according to the manufacturer's instructions. Briefly, the dipstick was immersed into the urine sample until all reagent squares were fully immersed. It was then immediately removed. The colour change on the reagent square due to oxidation, catalyzed by hemoglobin in urine, was inter-preted qualitatively as haematuria or no haematuria using the urinalysis guide provided by manufacturer.

## Urine examination for *S. haematobium* eggs

On each of three days, a 10 ml urine sample from each of child was filtered through a 12μm pore polycarbonate filter (Sterlitech) using a 13 mm filtration chamber. The filter was exam-ined under a microscope for *S. haematobium* eggs [17]. A mean egg count was calculated from the three days' readings and used to classify the infections as negative if no *S. haematobium* eggs were found, light if 1–49 eggs were found or heavy if $\geqq$50 eggs were found [4].

## Urine analysis for IL-6 and IL-10

Day I urine samples from 165 randomly selected children were assayed for IL-6 using Invitro-gen Human IL-6 ELISA kit (analytical sensitivity = <0.05 pg/mL; Assay range = 0.064–10000 pg/mL) with 96-well plates pre-coated with human IL-6 antibodies according to manufactur-er's instructions. The assay for IL-10 was done 190 randomly selected samples using DIA-source Human IL-10 96-well plate ELISA kits (Analytical sensitivity = <0.05 pg/mL; Assay range = 0.064–5,000 pg/mL) according to the manufacturer's instructions. The plates were read at an absorbance of 450 nm using Infinite 200PRO reader.

### Urinary tract ultrasound examination

A random sample of 165 children was selected prior to urine examination for *S. haematobium* eggs and their urinary tracts examined for ultrasound-detectable morbidity by an experienced ultrasonographer using a portable convex sector scanner (SSD-500; Aloka, Tokyo, Japan). Total morbidity was scored according to the Niamey protocol [18]. If any kidney and/ or ureter dilatation was observed the child was asked to empty the bladder and come back for re-examination. Morbidity was based on changes in urinary bladder (shape, wall lesions, wall thickening, masses and pseudopolyps), ureter dilatation and renal pelvis dilatation. Total morbidity scores were arbitrarily graded as 0 = no morbidity, 1–5 = light morbidity, 6–10 = moderate morbidity and $\geq 11$ = severe morbidity.

### Statistical analysis

Statistical analyses of the data were performed using STATA (Version 12) software. Median ages between boys and girls or between groups were compared using MannWhitney U tests or Kruskal-Wallis test, as appropriate. Medians of *S. haematobium* eggs, IL6 and IL-10 were compared between groups using Mann-Whitney U test or Kruskal-Wallis test. Prevalences were compared using $X^2$-test or logistic regression. The correlation between cytokine levels and *S. haematobium* egg counts or ultrasound-detectable pathology levels was analysed using Kendall Tau. Diagnostic performance of haematuria, IL-6 and IL-10 was assessed using Kappa statics. In all tests, *p*-values less than 0.05 were regarded as statistically significant.

### Ethics approval and consent to participate

The study protocol was reviewed by and received approval from the University of Nairobi-Kenyatta National Hospital Ethics and Research Committee (Approval No. P110/03/2017).

## Results

### Study population

A total of 245 children, 106 (43.3%) boys and 139 (56.7%) were girls, were recruited in the present study. The overall median age of the children was 10 years. The boys were significantly older than girls with median ages 11 and 10 years, respectively (*p* = 0.003). The children were grouped into 3 age groups; 5–9-year-olds, 10–11-year-olds and 12–15-year olds with 85 (34.7%), 121 (49.4%) and 39 (15.9%) children, respectively.

### *S. haematobium* prevalence in relation to age, sex and ultrasound-detectable urinary tract morbidity

The overall prevalence of *S. haematobium* infection among the children was 36.3% (89/245). The prevalence of the infection was significantly higher in boys 50.9% (54/106) than in girls 25.2% (35/139) (*p*<0.001). The infection prevalences were 22.4% (19/85), 42.2% (51/121) and 48.7% (19/39) for the age groups 5–9 years, 10–11 years and 12–15 years, respectively, with a significant difference in prevalences between the groups ($X^2$ = 11.538; *p* = 0.003).

The overall prevalence of detectable pathology was 35.8% (59/165). The intensity of morbidity ranged from none to severe. Fig 1 shows photographs of urinary bladders of two boys with diffuse bladder wall thickening and bladder masses. The intensity of morbidity in relation to sex, age and *S. haematobium* infection intensity is shown in Table 1. Overall, a significantly higher proportion of boys had ultrasound-detectable morbidity than girls ($X^2$ = 18.0760; *p*<0.001). Similarly, a significantly higher proportion of children with infection had pathology

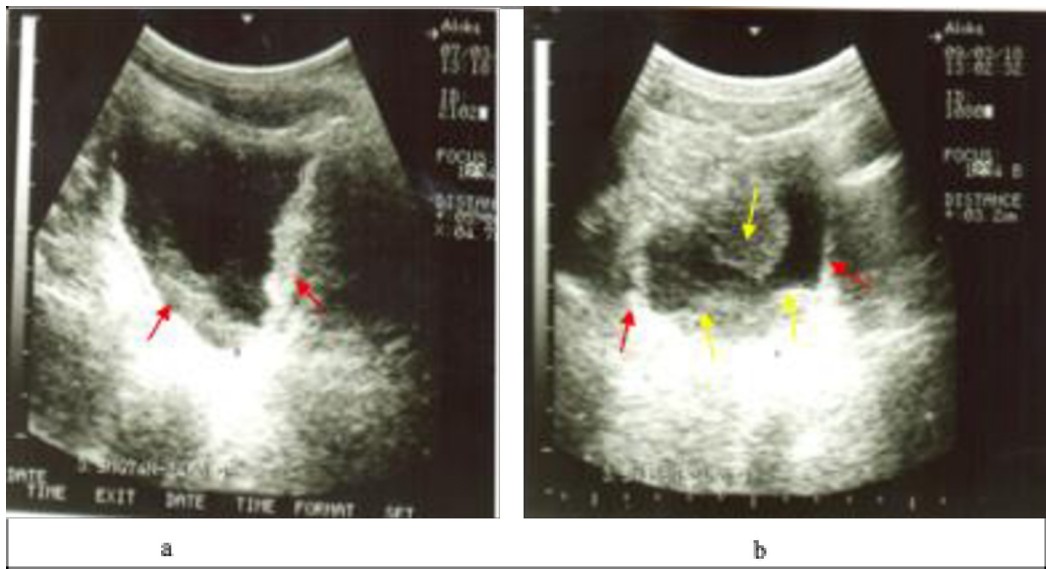

**Fig 1.** Urinary bladder pathology among the children; a) 10-year-old boy with diffuse thickening of the bladder wall (red arrows); b) 11-year-old boy with diffuse thickening of bladder wall (red arrows) and bladder masses (yellow arrows).

($X^2 = 14.3969$; $p < 0.002$). However, there was no significant difference in proportions of children with pathology in relation to age overall ($X^2 = 10.2826$; $p = 0.113$).

## Haematuria

Overall, the prevalence of haematuria among the children was 14.8% (n = 245). Among the boys, 23 (21.9%) had haematuria whereas 13 (9.4%) of the girls had haematuria. Girls had lower odds of having haematuria than boys (OR = 0.37, 95% CI: 0.18–0.77). This was statistically significant ($p < 0.008$). Age was significantly associated with prevalence of haematuria among the children where 6 (7.1%), 18 (15.0%) and 12 (30.1%) of those in age groups 5–9 years, 10–11 years and 12–15 years, respectively, had haematuria ($X^2 = 11.9610$; $p = 0.003$). In relation to infection intensity, 21 (60.0%) of children with heavy *S. haematobium* infection had haematuria whereas 7 (13.0%) of those with light infection and 8 (5.2%) of those with no infection had haematuria. This was statistically significant ($X^2 = 68.4476$; $p < 0.001$). The sensitivity and specificity of IL-6 and IL-10 ELISA and haematuria using urine microscopy as the reference are shown in Table 2. The performances of IL-6, IL-10 and haematuria were also plotted

**Table 1. Urinary tract morbidity in relation to sex, age and *S. haematobium* infection intensity.**

| Group | n | No morbidity | Light morbidity | Moderate morbidity | Severe morbidity | *p*-value |
|---|---|---|---|---|---|---|
| Boys | 72 | 47.2% | 25.0% | 18.1% | 9.7% | |
| Girls | 93 | 77.4% | 15.1% | 5.4% | 2.2% | <0.001 |
| 5–9 years old | 40 | 77.5% | 15.0% | 7.5% | 0.0% | |
| 10–11 years old | 89 | 62.9% | 16.9% | 11.2% | 9.0% | |
| 12–15 years old | 36 | 52.8% | 30.5% | 13.9% | 2.8% | 0.114 |
| Not infected | 105 | 74.2% | 16.2% | 6.7% | 2.9% | |
| Light Infection | 36 | 66.7% | 22.2% | 11.1% | 0.0% | |
| Heavy infection | 24 | 16.6% | 29.2% | 29.2% | 0.0% | <0.001 |
| **Overall** | **165** | **64.2%** | **19.4%** | **10.9%** | **5.5%** | |

**Table 2. Sensitivity and specificity of urinary IL-6, IL-10 and haematuria using microscopy as a reference.**

|  | Sensitivity | Specificity | PPV | NPV | EA | Kappa |
|---|---|---|---|---|---|---|
| IL-6 | 72.1% | 48.1% | 57.0% | 37.0% | 47.6% | 0.1796 |
| IL-10 | 86.1% | 22.9% | 46.8% | 37.9% | 42.6% | 0.0738 |
| Haematuria | 31.5% | 94.8% | 71.7% | 36.5% | 59.5% | 0.3012 |

PPV = Positive Predictive Value

NPV = Negative Predictive Value

EA = Expected agreement.

against that of urine microscopy. The area under the curve was largest for haematuria followed by IL-6 and IL-10 had the smallest area as shown in Fig 2.

## IL-6 in relation to sex and age

Urine samples from randomly selected 165 children were assayed for IL-6 levels. The prevalence of detectable levels of IL-6 in urine in relation to sex and age is shown in Table 3. Of the children tested, 98 (59.4%) had detectable levels of IL-6 in urine. There was no significant difference in the proportion of children with detectable levels of IL-6 in urine between boys and girls (n = 93) ($p$ = 0.301, OR = 0.72, 95%CI: 0.38–1.35).

There was no statistically significant difference in the levels of IL-6 in urine between boys and girls ($p$ = 0.535). The comparisons of urinary IL-6 levels between boys and girls, children with and without infection, children with and without haematuria and, children with and without urinary bladder are shown in Fig 3. In relation to age, there was a borderline statistical

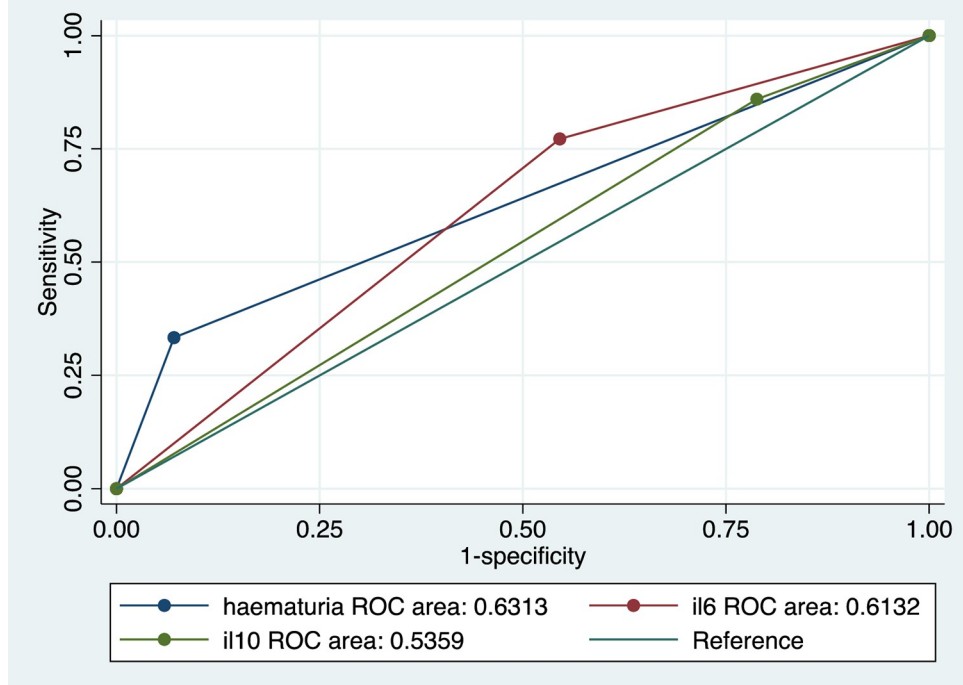

**Fig 2. ROC curve showing the areas under the curve for IL-6, IL-10 and haematuria using urine microscopy as the reference for the diagnosis of infection with *S. haematobium*.** il6 = IL-6, il10 = IL-10.

**Table 3. Association between IL-6 and gender, age, *S. haematobium* infection status, and haematuria.**

| Factor | Category | No. of children (n) | % with detectable IL-6 levels in urine | $X^2$ | *p*-value | Median (pg/ml) | *p*-value |
|---|---|---|---|---|---|---|---|
| **Gender** | Females | 72 | 55.9% | | 0.301 | 112.7 | |
| | Males | 93 | 63.9% | 1.0702 | | 129.0 | 0.550 |
| **Age (years)** | 5–9 | 54 | 72.2% | | 0.045 | 242.9 | |
| | 10–11 | 88 | 51.1% | | | 101.0 | |
| | 12–15 | 23 | 60.9% | 6.1935 | | 108.7 | 0.019 |
| ***S. haematobium* infection status** | Positive | 60 | 72.1% | | 0.011 | 157.2 | |
| | Negative | 105 | 51.9% | 6.5102 | | 107.2 | 0.086 |
| **Haematuria** | Present | 29 | 54.4% | | 0.005 | 106.7 | |
| | Absent | 136 | 82.8% | 7.9640 | | 108.9 | 0.021 |

difference in proportions with detectable IL-6 among the age groups and a statistically significant difference in the levels of IL-6 in urine among the age groups (*p* = 0.015).

## IL-10 in relation to sex and age

Urine samples from 190 children, of whom 83 were boys and 107 were girls, were assayed for IL-10 levels (Table 4). Of these children, 153 (80.5%) had detectable levels of IL10 in urine. There was no significant difference in the proportion of children with detectable levels of IL-10 in urine between boys (n = 83) and girls (n = 107) (*p* = 0.243, OR = 0.64, 95%CI: 0.30–1.35).

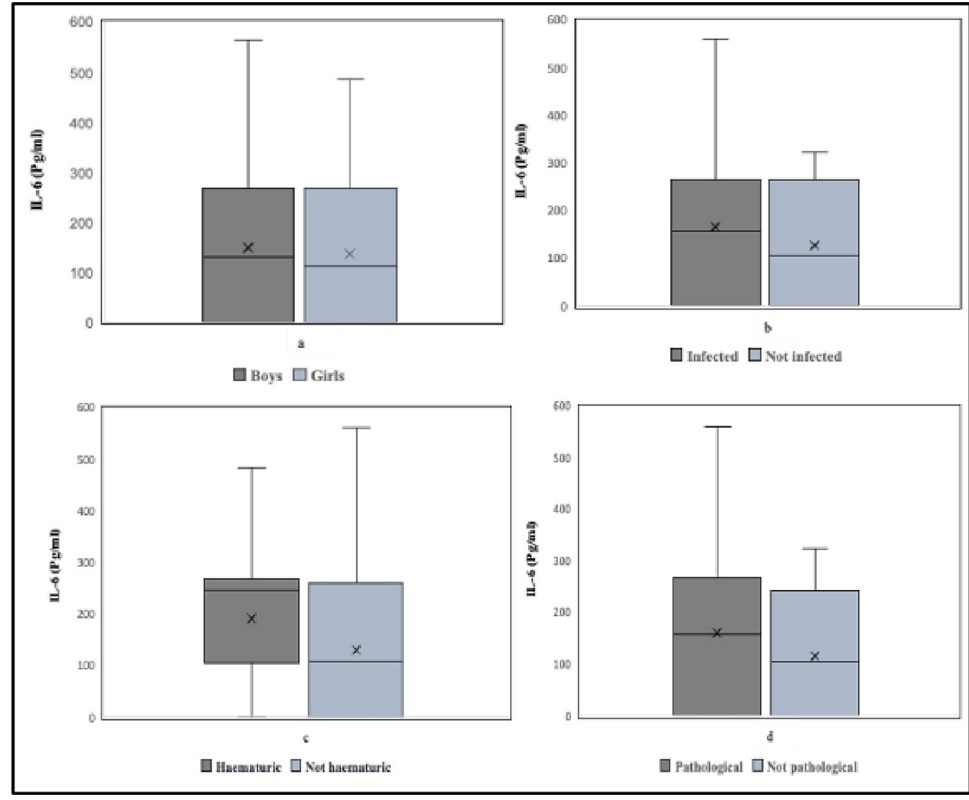

**Fig 3.** Box plot showing urinary IL-6 levels in relation to (a) sex, (b) *S. haematobium* infection, (c) haematuria and, (d) urinary bladder pathology. Solid line represents the median; X represents the mean and error bars represent the ranges.

**Table 4. Association between the prevalence of IL-10 and gender, age, *S. haematobium* infection status, and haematuria.**

| Factor | Category | No. of children (n) | % with detectable IL-10 in urine | $X^2$ | *p*-value | Median (pg/ml) | *p*-value |
|---|---|---|---|---|---|---|---|
| **Sex** | Females | 107 | 77.3% | | | 59.6 | |
| | Males | 83 | 84.3% | 1.3650 | 0.243 | 59.6 | 0.995 |
| **Age (years)** | 5–9 | 66 | 75.8% | | | 80.0 | |
| | 10–11 | 94 | 80.9% | | | 59.6 | |
| | 12–15 | 30 | 90.0% | 2.6805 | 0.262 | 59.6 | <0.001 |
| **_S. haematobium_ infection status** | Positive | 72 | 86.1% | | | 69.2 | |
| | Negative | 118 | 72.1% | 2.3059 | 0.129 | 59.6 | 0.301 |
| **Haematuria** | Present | 33 | 87.9% | | | 59.6 | |
| | Absent | 157 | 79.0% | 1.3767 | 0.241 | 59.6 | 0.216 |

There was no statistically significant difference in the levels of IL-10 in urine between boys and girls (*p* = 0.996). The comparisons of urinary IL-10 levels between boys and girls, children with and without *S. haematobium* infection, children with and without haematuria and, children with and without urinary balder pathology are shown in Fig 4. In relation to age, there was no statistical difference among the age groups but there was a statistically significant difference in the levels of IL-10 in urine among the age groups (*p*<0.001). There was no significant correlation between IL-10 and *S. haematobium* eggs (Tau = 0.047; *p* = 0.266).

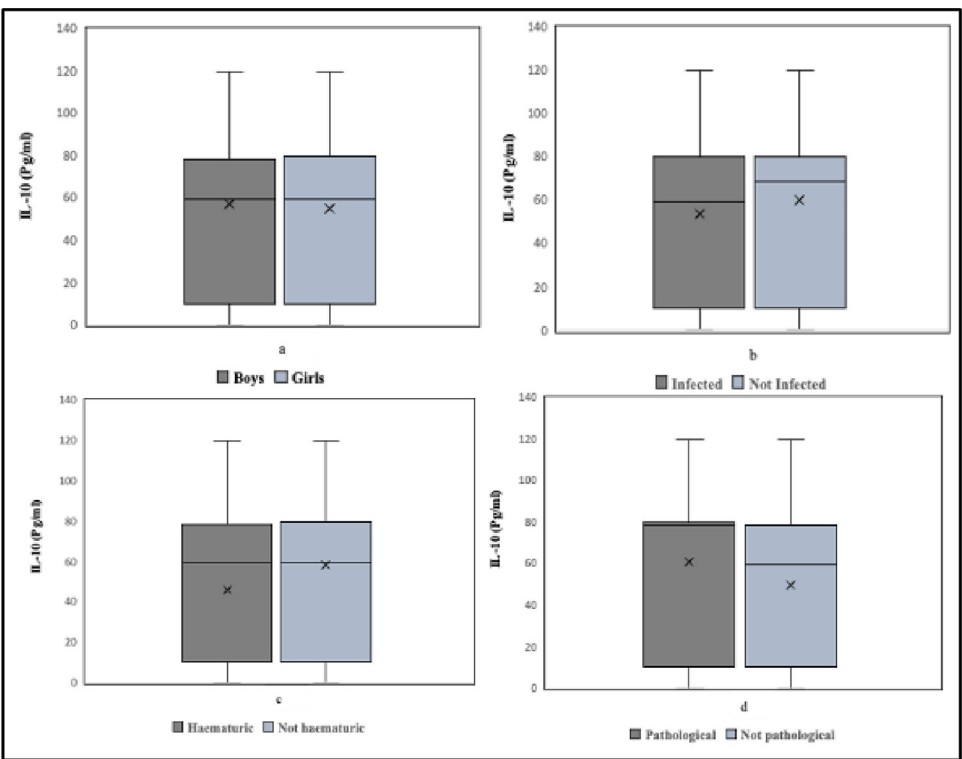

**Fig 4.** Box plot showing urinary IL-10 levels in relation to (a) sex, (b) *S. haematobium* infection, (c) haematuria and, (d) urinary balder pathology. Solid lines represent the median; X represents the mean and error bars represent the ranges.

### IL-6 and IL-10 in relation to morbidity

Of the 165 children examined for ultrasound-detectable urinary tract pathology, 112 also had their urine samples analysed for IL-6. Of these children, 37 (33.0%) had ultrasound-detectable pathology and 60 (53.6%) had detectable levels of IL-6 in urine. IL-6 levels were detected in urine samples from 39 (52.0%) of children without detectable pathology. The relationship between IL-6 levels and gender, age, infection status, haematuria and morbidity are shown in Fig 3. There was no significant correlation between IL-6 levels and ultrasound detectable pathology (Tau = 0.0794, $p$ = 0.322). There was no significant difference between those with and those without pathology ($X^2$ = 1.8916; $p$ = 0.635). Similarly, there was no significant statistical difference in IL-6 levels between children with and without ultrasound detectable pathology ($p$ = 0.384).

A random sample of 127 had both their urinary tracts examined for pathology and their urine samples analysed for IL10. Of these children, 43 (33.9%) had ultrasound-detectable pathology and 101 (79.5%) had detectable levels of IL-10 in urine. The relationship between IL-6 levels and gender, age, infection status, haematuria and morbidity are shown in Fig 4. Of those without detectable pathology 68 (81.0%) had detectable IL-10 levels in their urine whereas of those with detectable pathology 33 (76.7%) had detectable levels of IL-10. There was a negative but not significant correlation between IL-10 and ultrasound-detectable pathology levels (Tau = -0.0211; $p$ = 0.768). There was no significant difference in proportion between those with and those without pathology ($X^2$ = 1.3806; $p$ = 0.701). Similarly, there was no significant statistical difference in IL-10 levels between children with and without ultrasound detectable pathology ($p$ = 0.333).

### Effects of temperature on IL-6

The proportions of urine specimens with detectable levels of IL-6 after a 14-day storage at different temperatures were compared (Fig 5). A total 97 (59.4%) urine specimens had detectable levels of IL-6 after 14 days of storage at –20˚C. Of these, 61 (62.9%) also had detectable levels of IL-6 after storage at 4˚C for 14 days. Only 31 (31.6%) of urine specimens with detectable levels of IL-6 after storage at –20˚C also had detectable levels of IL-6 after storage at 25˚C. A total of 36.4% of urine specimens with detectable levels of IL-6 after storage at 4˚C also had detectable levels of IL-6 after storage at 25˚C. There were significant differences in IL-6 levels between urine specimens stored at –20˚C and those stored at 4˚C ($p$<0.001) and, between those stored at 4˚C and those stored at 25˚C ($p$<0.001).

### Effects of temperature on IL- 10

The proportions of urine specimens with detectable IL-10 after a 14-day storage at different temperatures were compared. A total of 38 (27.9%) of urine specimens with detectable levels of IL-10 after storage at –20˚C also had detectable levels of IL-10 after storage at 4˚C for 14 days. A total of 41 (31.5%) of urine specimens with detectable levels of IL-10 after storage at ⁻20˚C also had detectable levels of IL-10 after storage at 25˚C. A total of 17 (39.5%) of urine specimens with detectable levels of IL-10 after storage at 4˚C also had detectable levels of IL-10 after storage at 25˚C. There were significant differences in IL-10 levels between urine specimens stored at –20˚C and those stored at 4˚C ($p$<0.001) but not between those stored at 4˚C and those stored at 25˚C ($p$ = 0.871).

### Discussion

Urinary tract inflammation is associated with cytokines in urine which are increasingly being appreciated as biomarkers of urogenital pathology [6, 8–10, 19, 20]. A previous study in an

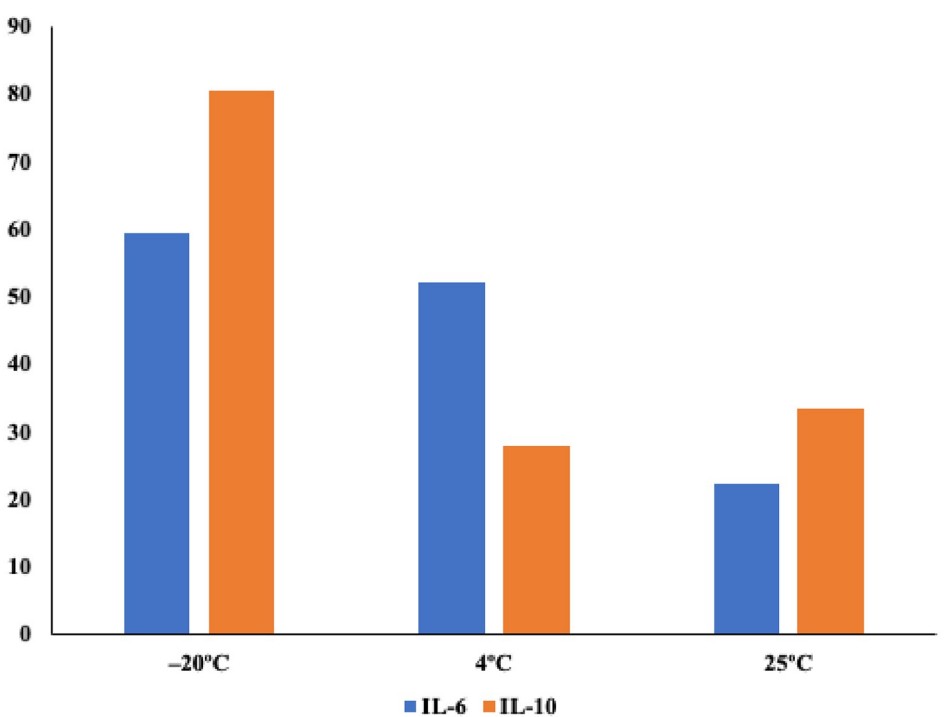

**Fig 5. Proportions of specimens with detectable levels of IL-6 or IL-10 after a 14-day storage at –20˚C, 4˚C or 25˚C.**

area highly endemic for *S. haematobium* suggested that urinary IL-6 and IL-10 have this potential but not much is known in relation to urinary cytokines and the influential factors in schistosomiasis [13]. The present study therefore assessed how urinary IL-6 and IL-10 are associated with age, sex and urinary tract pathology in children from an area with low *S. haematobium* prevalence.

The overall prevalence of *S. haematobium* infections (36.3%) was low compared to what was reported elsewhere in a similar study [13]. Boys had higher prevalence of infections and parasite egg counts than girls. However, both IL-6 and IL-10 in urine were not associated with sex of the children. No immediate explanation was available for this as it is expected that the sex with higher infections would have higher pro-inflammatory cytokines such as IL-6 and lower anti-inflammatory cytokines such as IL-10. It is however probable both boys and girls had the inflammation threshold to cause secretion of detectable cytokines in urine despite differences in shedding of parasite eggs in urine.

Urinary tract pathology was significantly higher in boys than in girls which corroborates the findings of a study in Angola [21]. However, in line with the association between sex and the cytokines, there were no significant differences in IL-6 and IL-10 between children with pathology and those without. Urinary cytokines are most likely secreted by activated immune cells in the urinary tract whereas ultrasound usually detects old pathology in which there could be minimal cellular activation and thus little cytokine secretion [22]. This could, at least partially, explain the lack of correlation between ultrasound-detectable pathology and urinary cytokines but requires further investigations. In addition, depending on the condition, cytokines such as IL-10 could have been produced by other non-inflammatory cells so that a huge proportion of the children had detectable levels regardless of infection status [23].

The prevalence and intensity of infections and increased with age. The prevalence of ultrasound detectable pathology also increased with age, although this was not significant, which corroborates the findings of a study in Sudan [24]. An increase in prevalence and levels of urinary IL-6 with age was also observed among the children. Taken together with the association between infection status and IL-6 levels, these suggest that inflammation processes continue with persistent infections as older children may have been exposed to infections for longer [25]. Eventually, immunomodulation kicks in and resulting in less pathology [26]. This could be the reason why children in the 10–11 years age group had higher morbidity and cytokine levels than those below or above them.

Our findings corroborate the findings of other studies that haematuria is a useful marker of *S. haematobium* infections in children [27–29]. Using urine microscopy as the reference, all three had poor agreement but kappa statistics showed that haematuria had higher agreement followed by IL-6 ELISA whereas IL-10 ELISA had the lowest. Low sensitivity of haematuria in low infections intensities have been reported elsewhere [28, 30]. Low infection prevalence and intensities could also be the reason for low sensitivity of urinary IL-6 and IL-10 but further studies should shed more light on this. In general these findings suggest that, similar to other urogenital schistosomiasis morbidity markers, the sensitivity of urinary cytokines may be reduced in areas with low infection prevalence and intensities [4].

Urine contains proteolytic enzymes that may denature proteins [31]. Urine storage temperatures, may therefore influence proteolytic degradation of urinary cytokines thus affecting their levels [32, 33]. There was observable reduction of proportion of specimens with detectable levels of IL-6 and IL-10 following a 14-day storage at 4˚C and 25˚C although above 4˚C there was less reduction for IL-10. These findings show that urine samples for cytokine assays should be frozen immediately after collection [34].

The present study only assayed a one-day urine specimen for cytokines which could be a potential limitation. Whereas previous studies have shown that schistosome egg shedding by infected individuals vary from day to day, no studies have been carried out on daily variation of urinary cytokine levels in relation to *S. haematobium* infections. Another limitation in the study is that after collection and exposure of the urine specimens to different temperatures, the specimens were stored at –80˚C for more than 8 weeks. During this period, due to the low temperatures, some of the vials broke or the lids came off and therefore such samples were not included in cytokine assays as there could have been possible contamination. However, this was deemed to be random and could not introduce any systemic bias in the results.

## Conclusions

The findings of the present study suggest that, unlike urinary IL-10, urinary IL-6 is associated with age, *S. haematobium* infection intensity and haematuria in children from low endemicity areas. However, it is not associated with ultrasound-detectable urinary tract morbidity. The results also suggest that both urinary IL-6 and IL-10 are significantly sensitive to storage temperatures above –20˚C. These findings call for further detailed studies to evaluate the potential of urinary IL-6 as *S. haematobium*-related morbidity marker.

## Supporting information

**S1 Database.**
(XLSX)

## Acknowledgments

We are grateful to the children from Duncan Ndegwa Primary School who agreed to participate in the study, and to the parents and teachers for allowing us to carry out the study in the school. Special thanks are extended to Charles Ng'ang'a and Jackson Muinde (Msambweni sub-District Hospital) for their unconditional support and participation during field sample collection. We are greatly indebted to Farah Bashir (KAVI-ICR) for his technical support and material contribution to this work. This study was funded by the European and Developing Countries Clinical Trials Partnership (EDCTP) for which we are greatly indebted.

## Author Contributions

**Conceptualization:** Kariuki H. Njaanake, Job Omondi.

**Formal analysis:** Kariuki H. Njaanake.

**Funding acquisition:** Kariuki H. Njaanake, Walter G. Jaoko, Omu Anzala.

**Investigation:** Kariuki H. Njaanake, Job Omondi, Omu Anzala.

**Methodology:** Kariuki H. Njaanake, Irene Mwangi, Omu Anzala.

**Project administration:** Kariuki H. Njaanake, Walter G. Jaoko, Omu Anzala.

**Supervision:** Walter G. Jaoko, Omu Anzala.

**Validation:** Kariuki H. Njaanake.

**Writing – original draft:** Kariuki H. Njaanake.

**Writing – review & editing:** Walter G. Jaoko.

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
