## [Decision Letter · Decision Letter 0]

28 Mar 2022

PGPH-D-22-00101

Urinary Interleukins (IL)-6 and IL-10 in Schoolchildren from an Area with low Prevalence of Urogenital Schistosomiasis in coastal Kenya

Dear Dr. Njaanake,

Thank you for submitting your manuscript to PLOS Global Public Health. After careful consideration, we feel that it has merit but does not fully meet PLOS Global Public Health’s publication criteria as it currently stands. Therefore, we invite you to submit a revised version of the manuscript that addresses the points raised during the review process.

We look forward to receiving your revised manuscript.

Kind regards,

Raquel Muñiz-Salazar, Ph.D.

Academic Editor

Journal Requirements:

1. We do not publish any copyright or trademark symbols that usually accompany proprietary names, eg (R), (C), or TM  (e.g. next to drug or reagent names). Therefore please remove all instances of trademark/copyright symbols throughout the text, including (R) on pages 6-7.

2. Your manuscript is missing the following sections: Introduction. Please ensure these are present, and in the correct order, and that any references to subheadings in your main text are correct. An outline of the required sections can be consulted in our submission guidelines here:

https://journals.plos.org/globalpublichealth/s/submission-guidelines#loc-parts-of-a-submission

3. Please provide  separate figure files in .tif or .eps format only and remove any figures embedded in your manuscript file.  Please ensure that all files are under our size limit of 20MB.  

For more information about how to convert your figure files please see our guidelines: Once you've converted your files to .tif or .eps, please also make sure that your figures meet our format requirements

4. In the online submission form, you indicated that "The data set analyzed for this manuscript will be availed, upon reasonable request, from the corresponding author.". All PLOS journals now require all data underlying the findings described in their manuscript to be freely available to other researchers, either 1. In a public repository, 2. Within the manuscript itself, or 3. Uploaded as supplementary information.

5. Please amend your detailed Financial Disclosure statement. This is published with the article, therefore should be completed in full sentences and contain the exact wording you wish to be published.

ii). State the initials, alongside each funding source, of each author to receive each grant.

iii). State what role the funders took in the study. If the funders had no role in your study, please state: “The funders had no role in study design, data collection and analysis, decision to publish, or preparation of the manuscript.”

iv). If any authors received a salary from any of your funders, please state which authors and which funders.

Additional Editor Comments (if provided):

Your manuscript has now been reviewed by experts in the field.

One of the significant issues is reinforcing the discussion and rewriting the conclusion.

Reviewers' comments:

Reviewer's Responses to Questions

**Comments to the Author**

1. Does this manuscript meet PLOS Global Public Health’s publication criteria? Is the manuscript technically sound, and do the data support the conclusions? The manuscript must describe methodologically and ethically rigorous research with conclusions that are appropriately drawn based on the data presented.

Reviewer #1: No

Reviewer #2: Yes

Reviewer #3: Yes

2. Has the statistical analysis been performed appropriately and rigorously?

Reviewer #1: No

Reviewer #2: Yes

Reviewer #3: Yes

3. Have the authors made all data underlying the findings in their manuscript fully available (please refer to the Data Availability Statement at the start of the manuscript PDF file)?

Reviewer #1: Yes

Reviewer #2: Yes

Reviewer #3: Yes

4. Is the manuscript presented in an intelligible fashion and written in standard English?

Reviewer #1: Yes

Reviewer #2: Yes

Reviewer #3: Yes

5. Review Comments to the Author

Reviewer #1: Aim of the study: The author designed the study to assay for levels of urinary cytokine Il-6 and Il-10 to be used used as diagnostic biomarker for urogenital schistosomiasis.

The title is not concise and clear.

The draft is composed of long sentences.

The references used in the study to justify the use of levels of cytokines and chemokine is far fetched since bacterial colonization in the urogenital organs and resultant pathology is different from parasitic infections. In S. hematobium infection, the cytokines released are as result of adult worms migrating into the venules surrounding the organs of the pelvis. Where they lay eggs and subsequently penetrate the vessel wall, move towards the lumen of the bladder. Some of the eggs are sequestered in the pelvic organs like the urinary bladder, lower ureters, vagina, prostate glands, and seminal vessicles where they cause chronic inflammation. Therefore, the assay of IL-6 and IL-10 is an understatement. In this parasitic phenomenon we should assay for a panel of urinary cytokines and chemokines to weed out the most prominent ones.

The author mixed up issues by further going looking at the effect of storage temperature on cytokine titers. This should have been an optimization protocol in the lab.

The author did not show any cytokine graphs or images of ultrasound detected pathologies.

Reviewer #2: Kariuki H Njaanake et.al. investigated the urine parameters and assays for Urogenital Schistosomiasis in schoolchildren in Kenya. They investigate the association of cytokines/protein ( IL6, IL10, ECP) with the Schistosoma haematobium egg detection from urine. Also, the clinical association analysis was also investigated to show the correlation of IL-6 and morbidity levels. Overall, the quality of the study is sound and the writing is structurally prepared. To improve the quality of the work, these questions and suggestions might help.

Major comments

-Title

” Urinary Interleukins (IL)-6 and IL-10 in schoolchildren from an area

with Low Prevalence of Urogenital Schistosomiasis in coastal Kenya”.

Why the low prevalence region in coastal Kenya is the appropriate site of study?, please clarify or add this clarified info somewhere in the paper.

-Abstract

“However, it is highly sensitive to urine storage temperatures”, this sentence interrupts the flow of reading.

Please exclude this, or add the result of temperature and IL-6 in the result section.

Method

- What is the inclusion /exclusion criteria of the participants?

- Authors analyzed the effect of temperature on IL-6in urine. How about the 2 consecutive urine samples were analyzed and interpreted. This could provide more insight into the IL-6 variation.

- How is the level of confidence to assess the level of hematuria from Urisxan Dipsticks. Is this the limitation?

- How the author confirms the result of S. haematobium eggs microscopic detection from the urine samples. Is this the limitation?

- -Why not all samples were tested for ECP (n=164), IL-6 (n=165) and IL-10 (n=190).

- Redundant info of ethical approval.

Result

Figure 1, how many times the author investigate the temperature effect to IL-6/IL10. I should be the error bar and statistical analysis.

Discussion

-Why aged group 10-11 has higher severe morbidity

-please add the rationale to focus only IL-10 and IL-6, why not other cytokines.

-Why specificity of ECP, IL6 IL10 and haematuria are so low.

-The part of cytokines IL6 IL10 seems not relevant to the main objective. Two options 1. Keep it, but need to add some info to link this into the main text, this can make the study more beneficial to the readers 2. Exclude to make the study concise and focus.

- Conclusion can be rewritten. Some info should be in the discussion rather than the conclusion.

Minor comment

-Title of table 2 should be revised, it is not only the sensitivity.

Reviewer #3: The following are the correction:

1. The title is not appropriate with the objectiveof the study. The study has shown the relationship with the IL-6 and IL-10 with the detection of S. haematobium egg. In that circumstances, the study will be design as diagnostic validity of IL-6 ad IL-10. Please explain.

2. The methodology section of abstract is missing with the study design, study place, study period. It should be mentioned in this section.

3. The introdction is nicely written. but the rationale of the study is missing.

4. The methodology section is nicely written. The details of the ELISA procedure can be avoided. No need to elaborate the procedure. Please mention "Th ELISA was performed accoridng to the manufacturer's instruction."

5. The results section should be written according to the objective of the study. The relevant table should be mentioned.

6. Rewrite the conclusion related with the findings of the study.

6. PLOS authors have the option to publish the peer review history of their article (what does this mean?). If published, this will include your full peer review and any attached files.

**Do you want your identity to be public for this peer review?** For information about this choice, including consent withdrawal, please see our Privacy Policy.

Reviewer #1: **Yes: **Nyamongo Onkoba, PhD

Reviewer #2: No

Reviewer #3: **Yes: **Dr. Md. Abdullah Yusuf

---

## [Decision Letter · Decision Letter 1]

14 Sep 2022

PGPH-D-22-00101R1

Urinary Interleukins (IL)-6 and IL-10 in Schoolchildren from an Area with low Prevalence of Schistosoma haematobium infections in coastal Kenya

Dear Dr. Njaanake,

Thank you for submitting your manuscript to PLOS Global Public Health. After careful consideration, we feel that it has merit but does not fully meet PLOS Global Public Health’s publication criteria as it currently stands. Therefore, we invite you to submit a revised version of the manuscript that addresses the points raised during the review process.

We look forward to receiving your revised manuscript.

Kind regards,

Raquel Muñiz-Salazar, Ph.D.

Academic Editor

Journal Requirements:

Additional Editor Comments (if provided):

The authors have improved the manuscript after addressing all reviewers’ comments. However, the manuscript still has significant drawbacks in terms of quality of writing and data and results from the presentation.

Major and minor comments are provided below. They should be addressed before this manuscript can be accepted for publication.

Reviewers' comments:

Reviewer's Responses to Questions

**Comments to the Author**

1. If the authors have adequately addressed your comments raised in a previous round of review and you feel that this manuscript is now acceptable for publication, you may indicate that here to bypass the “Comments to the Author” section, enter your conflict of interest statement in the “Confidential to Editor” section, and submit your "Accept" recommendation.

Reviewer #4: (No Response)

Reviewer #5: All comments have been addressed

Reviewer #6: (No Response)

2. Does this manuscript meet PLOS Global Public Health’s publication criteria? Is the manuscript technically sound, and do the data support the conclusions? The manuscript must describe methodologically and ethically rigorous research with conclusions that are appropriately drawn based on the data presented.

Reviewer #4: No

Reviewer #5: Yes

Reviewer #6: No

3. Has the statistical analysis been performed appropriately and rigorously?

Reviewer #4: Yes

Reviewer #5: Yes

Reviewer #6: No

4. Have the authors made all data underlying the findings in their manuscript fully available (please refer to the Data Availability Statement at the start of the manuscript PDF file)?

Reviewer #4: Yes

Reviewer #5: Yes

Reviewer #6: Yes

5. Is the manuscript presented in an intelligible fashion and written in standard English?

Reviewer #4: Yes

Reviewer #5: Yes

Reviewer #6: No

6. Review Comments to the Author

Reviewer #4: I was not one of the initial reviewers for this so will have different comments than those that were previously addressed.

The stated objective of the paper was to evaluate the relationship of IL-6 and IL-10 to pathology associated with urinary schistosomiasis. While there was an association of IL-6 with hematuria, neither cytokine showed any association with ultrasound-detectable pathology.

--on page 12 and 13, the authors mention that boys have a higher prevalence of infection and hematuria, respectively, yet the OR given was < 1.

--the quality of the figures is poor, especially figures 1 and 4. This may be an issue with the submission system but as seen in the reviewed material, they are not suitable for publication.

--the figure legends are inadequate. It should be possible to understand what is being shown and the significant findings without referring to the text. In figures 2 and 3, what does the bar represent? what does the "x" represent? what do the error bars represent? Do any of the panels represent statistical differences?

--figure 1 should include a panel from someone with no pathology to better show the contrast.

--figure 4 would be better as the overall concentration of IL-6 and IL-10 at the different time points although this experiment is somewhat unnecessary and it is well expected that storage at the wrong temperature would result in loss. It is true that transporting samples from the field cannot guarantee -20 during transport but any location that has an ELISA reader can reasonably be expected to have a freezer. At most, the experiment on getting the urines to a more appropriate temperature should be ~24 hours rather than 2 weeks.

--what are the normal ranges of IL-6 and IL-10 and what is considered abnormal? The authors seemed to have used "detectable" cytokine as the cutoff for tables 2, 3, and 4. This is not informative without knowing the limit of detection of the assays.

--references should be numbered in the text and bibliography

Reviewer #5: see attached

Reviewer #6: This manuscript investigated the association between urinary cytokine levels (IL6 & IL10) with the prevalence, intensity and morbidity of S. haematobium infection among schoolchildren in Kenya.

I've been given this manuscript as a revised (R1) version and I noticed that the manuscript has been improved after addressing reviewers’ comments given during previous reviewing stage. However, the manuscript still has major drawbacks in quality of writing and data & results presentation. Major and minor comments are provided below. They should be addressed before this manuscript can be acceptable for publication.

Note: A list of other major and minor comments is provided in the attached file.

7. PLOS authors have the option to publish the peer review history of their article (what does this mean?). If published, this will include your full peer review and any attached files.

**Do you want your identity to be public for this peer review?** For information about this choice, including consent withdrawal, please see our Privacy Policy.

Reviewer #4: No

Reviewer #5: No

Reviewer #6: **Yes: **Hesham M. Al-Mekhlafi

---

## [Decision Letter · Decision Letter 2]

27 Feb 2023

Urinary Interleukins (IL)-6 and IL-10 in Schoolchildren from an Area with low Prevalence of Schistosoma haematobium infections in coastal Kenya

PGPH-D-22-00101R2

Dear Dr. Njaanake,

We are pleased to inform you that your manuscript 'Urinary Interleukins (IL)-6 and IL-10 in Schoolchildren from an Area with low Prevalence of Schistosoma haematobium infections in coastal Kenya' has been provisionally accepted for publication in PLOS Global Public Health.

Best regards,

Julia Robinson

Executive Editor, PLOS Global Public Health

Please review the final comments from the two reviewers.

Reviewer Comments (if any, and for reference):

Reviewer's Responses to Questions

**Comments to the Author**

1. If the authors have adequately addressed your comments raised in a previous round of review and you feel that this manuscript is now acceptable for publication, you may indicate that here to bypass the “Comments to the Author” section, enter your conflict of interest statement in the “Confidential to Editor” section, and submit your "Accept" recommendation.

Reviewer #7: All comments have been addressed

Reviewer #8: All comments have been addressed

2. Does this manuscript meet PLOS Global Public Health’s publication criteria? Is the manuscript technically sound, and do the data support the conclusions? The manuscript must describe methodologically and ethically rigorous research with conclusions that are appropriately drawn based on the data presented.

Reviewer #7: Yes

Reviewer #8: Yes

3. Has the statistical analysis been performed appropriately and rigorously?

Reviewer #7: Yes

Reviewer #8: Yes

4. Have the authors made all data underlying the findings in their manuscript fully available (please refer to the Data Availability Statement at the start of the manuscript PDF file)?

Reviewer #7: Yes

Reviewer #8: Yes

5. Is the manuscript presented in an intelligible fashion and written in standard English?

Reviewer #7: Yes

Reviewer #8: Yes

6. Review Comments to the Author

Reviewer #7: This is an interesting article covering an important area of clinical symptoms / morbidity related to urogenital schistosomiasis. As national programs make progress in reducing prevalence and intensity of infection through chemotherapeutic control (and other measures) in line with the WHO’s 2030 roadmap, increasingly focus will turn to control and elimination of related morbidity.

An early stage test for schistosomiasis-related morbidity would be especially useful.

I found this article to be well-written, clearly laid out and methodologically sound. I was invited to review at the second revision stage and I note from earlier versions that the authors have made substantial changes to the manuscript. I only have some minor questions and assuming those are answered am happy to recommend it for publication.

Comments

• This is a low prevalence area as the authors state. Is it low infection at baseline (endemic equilibrium) or following treatment? Would this have an impact on the study?

• Would it be useful to also test in adults? Why was it restricted to children?

• Why is there a drop off in sample size from 245 to 165 children for ultrasound, and then to 112 for IL6

• The authors use gender and sex interchangeably. I’d suggest keeping consistent to sex.

Reviewer #8: The statememnt on page 5: The study was carried out in 2018 at Duncan Ndegwa primary school in Kwale County.

Page 6: The word is oxidation?

Throughout the discription in the manuscript it is rather appropriate to replace sex with gender.

Thank you for addressing all the comments as raised in the previous comments by the reviewers.

7. PLOS authors have the option to publish the peer review history of their article (what does this mean?). If published, this will include your full peer review and any attached files.

**Do you want your identity to be public for this peer review?** For information about this choice, including consent withdrawal, please see our Privacy Policy.

Reviewer #7: **Yes: **Michael French

Reviewer #8: **Yes: **Prof. Takafira Mduluza
